## Protocol for the development of the STrengthening the Reporting Of Pharmacogenetic Studies (STROPS) guideline: checklist of items for reporting pharmacogenetic studies

Marty Richardson,[1] Jamie J Kirkham,[1] Kerry M Dwan,[2] Derek J Sloan,[3] Geraint Davies,[4] Andrea Jorgensen[1]

[1]Department of Biostatistics, University of Liverpool, Liverpool, UK
[2]Cochrane Editorial Unit, London, UK
[3]Infection and Global Health Division, School of Medicine, University of Saint Andrews, Saint Andrews, Fife, UK
[4]Department of Clinical Infection, Microbiology and Immunology, University of Liverpool, Liverpool, UK

**Correspondence to**
Marty Richardson;
mhr@liverpool.ac.uk

## ABSTRACT

**Introduction** Large sample sizes are often required to detect statistically significant associations between pharmacogenetic markers and treatment response. Meta-analysis may be performed to synthesise data from several studies, increasing sample size and consequently power to detect significant genetic effects. However, performing robust synthesis of data from pharmacogenetic studies is often challenging due to poor reporting of key data in study reports. There is currently no guideline for the reporting of pharmacogenetic studies. The aim of this project is to develop the STrengthening the Reporting Of Pharmacogenetic Studies (STROPS) guideline. The STROPS guideline will facilitate the conduct of high-quality meta-analyses and thus improve the power to detect genetic associations.

**Methods and analysis** We will establish a preliminary checklist of reporting items to be considered for inclusion in the guideline. We will then conduct a Delphi survey of key stakeholder groups to gain consensus opinion on which reporting items to include in the final guideline. The Delphi survey will consist of two rounds: the first round will invite participants to score items from the preliminary checklist and to suggest additional relevant items; the second round will provide feedback from the previous round and invite participants to re-score the items. Following the second round, we will summarise the distribution of scores for each item, stratified by stakeholder group. The Steering Committee for the project and representatives from the key stakeholder groups will meet to consider the results of the Delphi survey and to finalise the list of reporting items. We will then draft, pilot-test and publish the STROPS reporting guideline and accompanying explanatory document.

**Ethics and dissemination** The University of Liverpool Ethics Committee has confirmed ethical approval for this study (reference: 3586). Dissemination activities will include presenting the reporting guideline at conferences relevant to pharmacogenetic research.

## INTRODUCTION

Pharmacogenetic studies investigate associations between genetic variants and treatment response for a particular drug of interest, in terms of both benefits (therapeutic effect) and harms (adverse effects). The aim of performing such studies is to identify ways that drug efficacy may be maximised, and that toxicity may be minimised. If a significant association between a genetic variant and a treatment response outcome is identified, patients may eventually be genotyped in clinical practice before being prescribed a certain treatment. The healthcare provider may then refer to the results of the genotyping test when determining whether to prescribe the drug, and if prescribed, the appropriate dosage of the drug. Such an approach is known as 'personalised medicine'.

Outcomes from pharmacogenetic studies are often likely to be complex traits; genetic influence may be explained by several genetic variants each having only a small effect on outcome. Consequently, large sample sizes are typically required to detect statistically significant associations between a genetic variant and treatment response. Meta-analysis improves sample size and consequently increases the power to detect significant

**Strengths and limitations of this study**

► We will conduct our project using methodology proposed by the EQUATOR (Enhancing the QUAlity and Transparency Of health Research) network for the development of reporting guidelines.
► The Delphi survey will enable us to gain information about the opinions of a wide group of participants.
► Our study design is limited by the fact that the consensus meeting will involve only the six members of the Steering Committee and one or two representatives from the key stakeholder groups and will be conducted via conference call.

genetic effects. However, significant differences are often observed between pharmacogenetic studies in terms of the genetic variants investigated, definition of genetic subgroups and outcomes, and assumptions made in the analyses for example about the underlying mode of inheritance. This can significantly reduce the number of studies available to contribute to a single meta-analysis. This problem is compounded by poor reporting of key data in study reports. For example, if the authors of a particular study do not report the number of participants in each genotype group and outcomes for each genotype group separately, it may not be possible for researchers conducting a systematic review to include this study in a meta-analysis. Furthermore, lack of reporting of participants' ethnic backgrounds can also severely hinder investigations of heterogeneity, which form a key part of any systematic review and/or meta-analysis. Genetic associations are likely to vary according to ethnicity; it is therefore recommended that meta-analyses are always stratified by ethnicity, and pooling of results should only be performed if effect estimates for different ethnic groups appear sufficiently similar.[1]

The aim of our project is to develop a guideline for the reporting of pharmacogenetic studies and an explanatory document using methodology proposed by EQUATOR (Enhancing the QUAlity and Transparency Of health Research).[2] Such a guideline would facilitate the conduct of high-quality systematic reviews and meta-analyses, thus improving power to detect genetic associations.

### An explanation of the terms used in this proposal
#### Steering Committee
The panel consists of six members: Marty Richardson (researcher into meta-analysis of pharmacogenetic studies), Jamie Kirkham (researcher into consensus methodology and developer of reporting guidelines), Kerry Dwan (researcher into systematic review methodology), Derek Sloan (Senior Clinical Lecturer in Infectious Diseases), Gerry Davies (Professor of Infection Pharmacology) and Andrea Jorgensen (researcher into statistical methods for pharmacogenetics, including evidence synthesis methods).

#### Delphi participants
Participants in the Delphi survey.

### METHODS AND ANALYSIS
We (the Steering Committee) will develop the guideline for pharmacogenetic studies in five stages:
1. Establish a preliminary checklist of reporting items to be considered for inclusion in the reporting guideline for pharmacogenetic studies (stage 1).
2. Conduct a Delphi survey to gain consensus opinion on reporting items to be considered within a reporting guideline for pharmacogenetic studies (stage 2).
3. Hold a consensus meeting to consider the results of the Delphi survey and to finalise the list of items for the reporting guideline (stage 3).

4. Develop and publish a high-quality reporting guideline and a detailed explanatory document (stage 4).
5. Dissemination activities to raise awareness of the published guideline, including presenting the guideline at relevant conferences (stage 5).

### Stage 1: Preliminary checklist of reporting items
We will establish a preliminary checklist of reporting items by:
i. Including items from existing relevant guidelines: Existing relevant guidelines will be identified by considering all guidelines listed on the EQUATOR website,[3] under the clinical area of genetics. Two authors (MR and ALJ) will assess existing guidelines to be relevant if they are applicable to pharmacogenetic studies from their experience of reviewing and undertaking meta-analyses of pharmacogenetic studies. We will include items from these existing guidelines in our preliminary checklist if they ensure transparency of reporting of pharmacogenetic studies and consequently will enable future evidence synthesis.
ii. Supplementing this list with any additional items thought to be important: Additional items will be identified by consideration of the quality assessment checklist for pharmacogenetic studies developed by Jorgensen and Williamson,[4] and through discussion among the Steering Committee.
iii. Providing an explanation and/or example of each reporting item: One author (MR) will draft the explanation and/or example for each reporting item. This will ensure that each reporting item is clear to participants in the Delphi survey.

All members of the Steering Committee will review and approve this preliminary list of reporting items and explanations/examples before the Delphi survey begins.

### Stage 2: Delphi survey
#### Design
The Delphi process will consist of two rounds of electronic-based survey, response and feedback. The first round survey will include scoring of reporting guideline items from the preliminary list formed at stage 1 and will invite additional items not included in this list. A second round survey will then be undertaken providing feedback from the previous round and inviting participants to re-score these items. Any additional reporting items identified by participants in the first round will be included for scoring by participants in the second round of the Delphi process.

#### Participants
We will invite three groups of stakeholders to participate in the Delphi survey. Stakeholder groups will be chosen to encompass all aspects of pharmacogenetic research.
1. Those who undertake primary pharmacogenetic research

We will ask members of pharmacogenetic networks, such as the Pharmacogenomics Research Network (PGRN) and the UK Pharmacogenetics and Stratified

Medicine Network, to participate in our Delphi survey. We will perform searches to identify these networks and contact experts in the field to ensure that all major networks across the globe are identified.

2. Those who systematically review pharmacogenetic research data

We will email the contact authors of systematic reviews of pharmacogenetic studies identified by searching PubMed, using appropriate search terms such as 'pharmacogenetics', 'pharmacogenomics', 'systematic review' and 'meta-analysis'. We will consult an information specialist (Eleanor Kotas) to design the search strategy. We will ask these authors to complete the survey if they participated in the data extraction and/or data synthesis of the review, and we will also ask this author to forward the survey on to other authors who participated in the data extraction and/or data synthesis.

3. Those who publish pharmacogenetic research

We will identify these individuals by contacting the editors-in-chief of key pharmacogenetic journals. We will perform searches to identify key journals, using search terms 'pharmacogenetics', 'pharmacogenomics', 'precision medicine', 'personalised/personalized medicine'. We will also consider journals listed on the 'The SCImago Journal & Country Rank' portal[5] under the subject category 'Genetics'. We will contact experts in the field to ensure that we have not missed any key journals. We will ask the editors-in-chief to participate in the survey themselves and also to forward the survey on to editors at their journal.

The decision of how many individuals to invite to complete a Delphi survey is not based on statistical power and often must be a pragmatic choice.[6] Generally, there should be good representation of experts from the key stakeholder groups who have a deep understanding of the relevant issues. Our aim is to maximise the number of participants who complete the Delphi survey to ensure that the total number of participants within each stakeholder group is sufficiently large to yield a meaningful statistical analysis.

For the stakeholder group of primary researchers, we are confident that a sufficiently large number of individuals will complete the survey; two of the networks (the PGRN network and the UK Pharmacogenetics and Stratified Medicine Network) whose members we will invite to complete the survey together have over 1000 members. For the stakeholder groups of systematic reviewers and journal editors, we hope that the snowballing technique, that is asking review authors and editors-in-chief to forward the survey on to other individuals, will ensure that the number of participants in these groups is sufficiently large.

## Recruitment process and ethical considerations

We will email the individuals listed above with information about the STROPS project and the Delphi process and an invitation to complete round 1 of the Delphi survey within 3 weeks (see online supplementary file 1). We will inform invitees that participation in the survey is optional and that we will assume informed consent if an invitee responds to round 1 of the survey. We will inform invitees that all data will be anonymised, and we will allocate a unique identification number to each participant in the Delphi survey. When registering for the study on the survey website, participants will also be asked to tick a box if they agree to participate in the study.

We will send a reminder email at the end of the second week to prompt completion of the survey. We will not be able to send a reminder email to individuals who received forwarded invitations, as we will not have contact information for these individuals at this stage. All participants who complete the first round of the Delphi survey will be invited to participate in the second round. However, we will inform invitees that completion in the first round does not necessitate completion in the second round, and we will remind participants of the first round that this is the case, when we invite them to complete the second round.

If attrition rates (the degree of non-response to the second round of the survey) are high, either for a particular stakeholder group or overall, then we will adopt strategies to increase response rates. Generally, a response rate of around 80% for the second round of a Delphi survey can be considered satisfactory in most scenarios.[6] Strategies for increasing response rates to round 2 may include sending personalised reminder emails, offering acknowledgement in the published reporting guideline and extending the period of time for which the second round is open.

If participants who do not respond to round 2 have different opinions to participants from the same stakeholder group who complete both rounds, then attrition bias has occurred and this will affect the results of the Delphi survey. If response rates to round 2 are less than 80%, we will investigate the risk of attrition bias. We will calculate average round 1 scores for each participant, and then plot these scores according to whether participants completed round 2 or not for each stakeholder group. We will visually examine these plots to assess the likelihood of attrition bias.

## Participant characteristics

We will ask participants to provide their name, email address and their consent to be acknowledged as a participant in the Delphi survey in publications arising from this project. Demographic data regarding the participant's profession and previous involvement with reporting guideline development will be collected; all demographic data will be anonymised.

## Delphi scoring

Participants will be asked to score each of the reporting guideline items listed using a scale of 1 to 9, with 1 to 3 labelled 'not important for inclusion in the guideline', 4 to 6 labelled 'important but not critical for inclusion in the guideline' and 7 to 9 labelled 'critical for inclusion into the guideline'.[7] Participants will also be given the

option to score a reporting guideline item as 'unable to score' if they are unable to offer an opinion as to whether the item is important or not.

### Software

The Delphi survey will be conducted using DelphiManager, a web-based system designed by the COMET Initiative (http://www.comet-initiative.org/delphimanager/) to facilitate the building and management of Delphi surveys.

### Delphi round 1

Reporting guideline items will be presented in the order in which they would be addressed in the pharmacogenetic study report and will be grouped under relevant headings (ie, title and abstract, introduction, methods, results, discussion and other information). Participants will be asked to score each item as described previously. Participants will also be given the chance to add items that they believe should be included in the reporting guideline.

### Round 1 analysis

For each item, the number of participants who have scored the item and the distribution of scores will be summarised. Participants who scored an item as 'unable to score' will be excluded from the analysis for that particular item. We will review all additional guideline reporting items listed by participants to ensure that they are not covered by the existing list of reporting guideline items. Additional reporting guidelines items that are not already covered will undergo formal review and discussion by the Steering Committee, and if appropriate, will be added to the list of reporting guideline items presented in round 2.

### Delphi round 2

In round 2, each participant who participated in round 1 will be shown the number of respondents and distribution of scores for each item from round 1, for each stakeholder group separately. Participants will also be reminded how they personally scored each item in round 1. Participants will be asked to consider the responses from other Delphi participants and to re-score the items.

In addition, if a participant changes their score from 'not critical' in round 1 to 'critical' in round 2 or from 'critical' in round 1 to 'not critical' in round 2, they will be asked to provide their reasoning for this change.

Additional items identified as part of round 1 will be scored by participants in round 2.

### Round 2 analysis

For each item, the number of respondents and the distribution of scores will be summarised. Participants who scored an item as 'unable to score' will be excluded from the analysis for that particular item. We will assess the possibility of attrition bias occurring by comparing item scores from participants who completed round 1 only, to item scores from participants who completed both round 1 and round 2. We will also examine changes in scores

between rounds and summarise the reasons given for changes from 'critical' to 'non-critical' and vice versa.

### Consensus definition

Guideline reporting items will be prioritised if at least 70% of participants score them as 'critical'. The rationale for this threshold is that consensus that an item ought to be included in the reporting guideline requires agreement by the majority regarding the critical importance of the outcome. This threshold for consensus has been used previously in the development of the COS-STAR reporting guideline.[8]

### Stage 3: Consensus meeting

The Steering Committee and one or two representatives from each stakeholder group will meet to consider the results of the Delphi survey and to finalise the list of items for the draft reporting guideline. Representatives from each stakeholder group must have completed both rounds of the Delphi survey. Furthermore, we aim to include at least one non-UK representative from each stakeholder group in the consensus meeting. The meeting will be conducted via conference call.

At the meeting, MR will present a summary of the results of how each stakeholder group had scored each reporting guideline item (from stage 2), and the number of stakeholder groups who achieved consensus. Meeting attendees will discuss each reporting guideline item in turn and will make a decision on whether to include the item in the reporting guideline or not. Items that reached consensus from all stakeholder groups in the Delphi survey will be considered first. Each remaining item will then be considered in turn according to the number of stakeholder groups where consensus was achieved, that is, the next batch of items to be considered will be those that reached consensus in all but one stakeholder group.

### Stage 4: Development and publication of reporting guideline and explanatory document

We will draft the initial reporting guideline and the explanatory document concurrently. The purpose of the explanatory document is to provide the meaning and rationale for each reporting item alongside examples of good reporting practice. For each item, we will also document the origin of the item (Steering Committee or Delphi participants) and the degree of consensus achieved from the Delphi survey.

We will pilot-test the draft reporting guideline with researchers who are yet to publish the findings of their pharmacogeneticstudy and with researchers who have already published a pharmacogenetic study. We will identify these individuals by contacting pharmacogenetic researchers based at the University of Liverpool and by contacting authors of published pharmacogenetic studies listed on PubMed. We will incorporate feedback on the content, format and usefulness of the guideline from these researchers in the final reporting guideline.

## Stage 5: Dissemination of the reporting guideline

Dissemination activities will include presenting the final reporting guideline at conferences relevant to pharmacogenetic research.

## Patient and public involvement statement

The STROPS guideline will be developed without patient involvement. Patients were not invited to contribute to the writing or editing of this document for readability or accuracy.

## DISCUSSION

We plan to conduct our project using robust methodology for developing reporting guidelines proposed by the EQUATOR network.[2] Using such methodology will ensure that the resulting reporting guideline is useful and widely disseminated. The EQUATOR approach includes a face-to-face consensus meeting, which follows the Delphi survey. This meeting often involves the steering group and a selection of stakeholders who took part in the Delphi survey. Simera *et al*[9] conducted a survey of authors of 37 reporting guidelines and reported that the median number of people participating in consensus meetings for these reporting guidelines was 22. Due to a lack of funding for this project to cover travel and accommodation costs, we will be unable to arrange a face-to-face consensus meeting including such a large number of participants. Our consensus meeting will only involve the members of the Steering Committee (n=6) and one or two representatives of the key stakeholder groups, and the meeting will be conducted via conference call. We will invite a large, international and multidisciplinary cohort to participate in the Delphi survey, so that meeting attendees are able to base their decisions on the opinions of this wider cohort.

We will prioritise items for inclusion in the guideline if at least 70% of participants score them as 'critical'. Although the choice of this threshold is somewhat subjective, pre-specification of the threshold in this protocol ought to provide assurance that we will not define consensus in a post-hoc way, and therefore our own opinions will not bias the results of the Delphi survey.[10]

There is currently no guideline for the reporting of pharmacogenetic studies that has been developed using a widely accepted robust methodology. The final guideline will not only improve the transparency of reporting of pharmacogenetic studies but also facilitate the conduct of high-quality systematic reviews and meta-analyses, and

thus improve the power to detect genetic associations. With the increasing number of meta-analyses of pharmacogenetic studies that are being undertaken, it is important that reporting of key data in study reports is improved in order to allow robust synthesis of the studies.

**Contributors** MR, AJ, JJK and KMD conceived the idea for the project. All authors (MR, AJ, JJK, KMD, GD and DJS) contributed to the design of the study. MR is the guarantor and drafted the manuscript. All authors (MR, AJ, JJK, KMD, GD and DJS) read, provided feedback and approved the final manuscript.

**Funding** MR is supported partly by Liverpool Reviews and Implementation Group (LRiG, based at the University of Liverpool), based on funding from the National Institute for Health Research Health Technology Assessment Programme (URL, http://www.nets.nihr.ac.uk/programmes/hta), and partly by the Research, Evidence and Development Initiative (READ-It) project. READ-It (project number 300342-104) is funded by UK aid from the UK government; however, the views expressed do not necessarily reflect the UK government's official policies.

**Competing interests** None declared.

**Patient consent for publication** Not required.

**Ethics approval** The University of Liverpool Ethics Committee has been consulted and confirmed ethical approval for this study (Reference: 3586).

**Provenance and peer review** Not commissioned; externally peer reviewed.

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
