## [Reviewer comments · BMJ Open]

ARTICLE DETAILS

TITLE (PROVISIONAL)	Protocol for the development of the STrengthening the Reporting Of Pharmacogenetic Studies (STROPS) guideline: checklist of items for reporting pharmacogenetic studies
AUTHORS	Richardson, Marty; Kirkham, Jamie; Dwan, Kerry M; Sloan, Derek; Davies, Geraint; Jorgensen, Andrea

VERSION 1 – REVIEW

REVIEWER	Ann Moyer Mayo Clinic, USA
REVIEW RETURNED	16-Mar-2019

GENERAL COMMENTS	Addressing the variability in pharmacogenetics studies by providing a guideline is a great idea. My primary concern with this protocol as written is that the scope is not clear. The authors state that there is "poor reporting of key data" but do not elaborate on what the "key data" may include (I have ideas of what I think the "key data" are, but am not sure if that aligns with what the authors were thinking). While the "key data" and scope will likely be in part defined by the Delphi process, given that participants can suggest "additional guideline reporting items," it would be helpful to include examples in the introduction to give the readers of this article a sense of what the study is attempting to address. A few minor questions: -The authors state that although a consensus meeting would typically include ~22 people, due to lack of funding, only the steering committee members will participate rather than including representatives of key stakeholder groups. Was a conference call considered as an option?-In the introduction, the authors state "Pharmacogenetic studies investigate associations between genetic variants (SNPs)...." They may wish to broaden this to genetic variation rather than variants (because haplotypes are common in PGx and there are limitations to studies that do not take the haplotype structure into account) and if they use SNVs as an example, they may wish to also include CNVs as well-Have the authors considered including the end-users of pharmacogenetics research as participants as well? When providers, clinical implementation groups, and laboratories are trying to make decisions about whether to offer/implement new genes/variants in the form of clinical tests, we also rely on the literature and the way studies are done and/or reported can present challenges. Therefore, these "end-users" may represent another group of stakeholders with a unique perspective that may
---

	be valuable when creating a guideline - particularly if the ultimate goal is bench-to-bedside translation. -under "3. Those who publish pharmacogenetics research" - a Pubmed search here (similar to that described in #2) may be helpful as well. Overall, I think this is important and interesting work and will be looking forward to seeing the results.
--	---

REVIEWER	Eberechukwu Onukwugha Associate Professor, University of Maryland School of Pharmacy USA
REVIEW RETURNED	02-Apr-2019

GENERAL COMMENTS	This manuscript provides the protocol for a study that will develop guidelines for pharmacogenetics studies. This is an area in which guidelines are critically lacking. The availability of guidelines would meet a recognized need in the field. Overall, the authors present a clear, easy-to-follow study protocol. The authors may want include individuals involved in growing areas, i.e., those who conduct comparative effectiveness studies and economic evaluations in the area of personalized medicine. I provide comments for the author's consideration in the following text. P4, L21-23: this may be understood to mean that the availability of guidelines is causally related to the improvement in reporting. Please rephrase to be consistent with the language on pg 6, L52-57. P7, L55: how is stage 5 to be distinguished from the publishing objective in stage 4? Provide more detail in stage 5. P9, L46-47: will you have the contact information for the individuals to whom your original invitees outreach? P10, L3-10: is the focus on maximizing the total number of participants or the total number within each group? If the latter, there could be tailored approaches for each stakeholder group that the team adopts to increase numbers, versus a one-size-fits-all approach to increase numbers for the total sample. What is the number that is "sufficiently large"? Is this number a target number or a minimum threshold? Does the same number apply to the Round 2 of the Delphi stage? P10, L5: add "total" between "the" and "number" if this is correct. P10, L22: please provide more detail regarding the informed consent process. It has not been described and thus it is difficult to determine whether it is appropriate to assume informed consent when an invitee responds. P10, L30-33: will the reminder communication also be provided to those who received forward invitations? (see comment for p9, L46-47). P11, L10-12: numerically, how will you handle "unsure" responses in the analysis phase? P12, L48: what is the justification for the choice of 70 percent?
--

REVIEWER	Jacqz-Aigrain Evelyne APHP - Hopital Robert Debré - Paris - France
REVIEW RETURNED	10-May-2019

GENERAL COMMENTS	The manuscript presents a project of Delphi study aiming to develop a checklist of key items to report pharmacogenetic studies. The Delphi process is described in details, and id planned according to "guidelines / recommendations". Comments and precisions The preliminary checklist is developed only by the Sterring Committee. Although the 5 members have recognized expertise, is there a possibility that participants propose new items during the first round ? It is usual to "recommend" that participants complete both rounds. What would be the impact of having a significant number of participants not responding to the second round ? Should the authors decide of a "kind of limit" in the number of participants to both rounds under which interpretation will have a bias ? Regarding the snowballing to ensure that the number of participants is large enough, how will their "qualification" to answer the Delphi be evaluated ? the expected number of participants might be pre-defined but more important,they should be selected based on their knowledge of the topic.
---

VERSION 1 – AUTHOR RESPONSE

Reviewer 1 Comments:

1. My primary concern with this protocol as written is that the scope is not clear. The authors state that there is "poor reporting of key data" but do not elaborate on what the "key data" may include (I have ideas of what I think the "key data" are, but am not sure if that aligns with what the authors were thinking). While the "key data" and scope will likely be in part defined by the Delphi process, given that participants can suggest "additional guideline reporting items," it would be helpful to include examples in the introduction to give the readers of this article a sense of what the study is attempting to address.

RESPONSE: We have added examples of "key data" in the introduction to the protocol (see tracked changes in the revised manuscript), to give readers of the article a sense of what the study is attempting to address.

2. The authors state that although a consensus meeting would typically include ~22 people, due to lack of funding, only the steering committee members will participate rather than including representatives of key stakeholder groups. Was a conference call considered as an option?

RESPONSE: We considered that a conference call involving such a large number of individuals would not be conducive to a successful meeting, due to a lack of spontaneity and limited interaction between participants. However, reflecting on this reviewer's comment, we have decided that a conference call involving the steering committee and one or two representatives from each stakeholder group would be feasible. We have edited the protocol to reflect this (see tracked changes in the abstract, article summary, methods and analysis, discussion).

3. In the introduction, the authors state "Pharmacogenetic studies investigate associations between genetic variants (SNPs)...." They may wish to broaden this to genetic variation rather than variants (because haplotypes are common in PGx and there are limitations to studies that do not take

the haplotype structure into account) and if they use SNVs as an example, they may wish to also include CNVs as well

RESPONSE: In the first paragraph of the introduction, we have removed the reference to SNPs as we do mean genetic variation in general, including haplotypes, rather than only SNPs. Thank you for pointing this out.

4. Have the authors considered including the end-users of pharmacogenetics research as participants as well? When providers, clinical implementation groups, and laboratories are trying to make decisions about whether to offer/implement new genes/variants in the form of clinical tests, we also rely on the literature and the way studies are done and/or reported can present challenges. Therefore, these "end-users" may represent another group of stakeholders with a unique perspective that may be valuable when creating a guideline - particularly if the ultimate goal is bench-to-bedside translation.

RESPONSE: Thank you for this suggestion. We have now opened the Delphi survey and so are unable to add stakeholder groups at this stage. However, we will look for "end-users" of pharmacogenetic research to invite to the consensus meeting, which has not yet taken place. If we are able to locate such individuals, we will report the invitation of representatives of this group as a protocol amendment when we publish the final reporting guideline.

5. Under "3. Those who publish pharmacogenetics research" - a Pubmed search here (similar to that described in #2) may be helpful as well.

RESPONSE: We have added detail on how we will identify individuals from the third stakeholder group under "3. Those who publish pharmacogenetics research" (see tracked changes in the revised manuscript).

Reviewer 2 Comments

1. P4, L21-23: this may be understood to mean that the availability of guidelines is causally related to the improvement in reporting. Please rephrase to be consistent with the language on pg 6, L52-57.

RESPONSE: We have edited this sentence to be consistent with the language used on pg 6, L52-57. We no longer say that the guideline will improve the reporting of pharmacogenetic studies (see tracked changes in the revised manuscript)

2. P7, L55: how is stage 5 to be distinguished from the publishing objective in stage 4? Provide more detail in stage 5.

RESPONSE: By "dissemination of the published guideline", we mean the process of raising awareness of the published reporting guideline, including presenting at relevant conferences. We have added detail to "Stage 5" to make this clear (see tracked changes in the revised manuscript).

3. P9, L46-47: will you have the contact information for the individuals to whom your original invitees outreach?

RESPONSE: When participants sign up for the Delphi survey, we will ask them to provide their name and email address, so we will have contact information for the individuals to whom the original

invitees outreach if they complete the survey. We have edited line 46 on page 10 to make this clearer (see tracked changes in the revised manuscript).

4. P10, L3-10: is the focus on maximizing the total number of participants or the total number within each group? If the latter, there could be tailored approaches for each stakeholder group that the team adopts to increase numbers, versus a one-size-fits-all approach to increase numbers for the total sample. What is the number that is “sufficiently large”? Is this number a target number or a minimum threshold? Does the same number apply to the Round 2 of the Delphi stage?

RESPONSE: The focus is on maximising the total number of participants within each group. We have edited the text so this is clear. We have also included details on the tailored approach we are taking to maximise the number of participants in each group. The decision of how many individuals to invite to complete a Delphi survey is not based on statistical power, and is often a pragmatic choice. We have therefore not set a target sample size. Generally, there should be good representation of experts from the key stakeholder groups, who have a deep understanding of the relevant issues. We have also included this detail (see tracked changes in the revised manuscript under the heading “Participants”).

5. P10, L5: add “total” between “the” and “number” if this is correct.

RESPONSE: The reviewer is correct, we have added “total” between “the” and “number”.

6. P10, L22: please provide more detail regarding the informed consent process. It has not been described and thus it is difficult to determine whether it is appropriate to assume informed consent when an invitee responds.

RESPONSE: We have provided more information on the informed consent process under the heading “Recruitment process and ethical considerations” (see tracked changes in the revised manuscript), including the addition of a supplementary file which contains the information that will be provided to invitees in the initial email. Please also note that the University of Liverpool Ethics Committee have reviewed all planned study processes (including the process of obtaining informed consent) and have confirmed ethical approval for this study.

7. P10, L30-33: will the reminder communication also be provided to those who received forward invitations? (see comment for p9, L46-47).

RESPONSE: We will not be able to send a reminder e-mail to individuals who received forwarded invitations, as we will not have contact information for these individuals at this stage. We have indicated this in the manuscript under the heading “Recruitment process and ethical considerations” (see tracked changes in the revised manuscript).

8. P11, L10-12: numerically, how will you handle “unsure” responses in the analysis phase?

RESPONSE: We have edited P11 L10 to say “unable to score” instead of “unsure”, as this is the option participants will be given. Participants who score an item as “unable to score” will be excluded from the analysis for that particular item. We have indicated this under the headings “Round 1 analysis” and “Round 2 analysis” (see tracked changes in the revised manuscript).

9. P12, L48: what is the justification for the choice of 70 percent?

RESPONSE:

The rationale for this threshold is that consensus that an item ought to be included in the reporting guideline requires agreement by the majority regarding the critical importance of the outcome. This

threshold for consensus has been used previously in the development of the COS-STAR reporting guideline. We have added this detail to P12, L48.

Although the choice of this threshold is somewhat subjective, pre-specification of the threshold in this protocol ought to provide assurance that we will not define consensus in a post-hoc way, and therefore our own opinions will not bias the results of the Delphi survey. We have added this detail to the discussion section of the protocol.

Reviewer 3 Comments:

1. The preliminary checklist is developed only by the Steering Committee. Although the 5 members have recognized expertise, is there a possibility that participants propose new items during the first round ?

RESPONSE: As stated on page 11 (L37-39), “participants will also be given the chance to add items that they believe should be included in the reporting guideline”.

2. It is usual to “recommend” that participants complete both rounds. What would be the impact of having a significant number of participants not responding to the second round ? Should the authors decide of a “kind of limit” in the number of participants to both rounds under which interpretation will have a bias ?

RESPONSE: For ethical reasons, we want to make it clear that participants are able to withdraw from the survey at any time. We therefore plan to inform participants that completion in the first round does not necessitate completion in the second round, and to remind participants of the first round that this is the case, when we invite them to complete the second round (see text under “Recruitment process and ethical considerations”).

If attrition rates (the degree of non-response to the second round of the survey) are high, either for a particular stakeholder group or overall, then we will adopt strategies to increase response rates. Generally, a response rate of around 80% for the second round of a Delphi survey can be considered satisfactory in most scenarios. Strategies for increasing response rates to Round 2 may include: sending personalised reminder e-mails, offering acknowledgement in the published reporting guideline, and extending the period of time for which the second round is open.

If participants that do not respond to Round 2 have different opinions to participants from the same stakeholder group who complete both rounds, then attrition bias has occurred and this will affect the results of the Delphi survey. If response rates to Round 2 are less than 80%, we will investigate the risk of attrition bias. We will calculate average Round 1 scores for each participant, and then plot these scores according to whether participants completed Round 2 or not for each stakeholder group. We will visually examine these plots to assess the likelihood of attrition bias. We have added this detail to the protocol (see tracked changes under “Recruitment process and ethical considerations”).

3. Regarding the snowballing to ensure that the number of participants is large enough, how will their “qualification” to answer the Delphi be evaluated? the expected number of participants might be pre-defined but more important, they should be selected based on their knowledge of the topic..

RESPONSE: The Delphi survey is substantial (we expect it will take approximately 60 minutes to complete) and requires careful consideration and knowledge of pharmacogenetics for an individual to be able to engage with the survey. We are therefore confident that no individuals who do not have expertise in the area of pharmacogenetics would respond to the survey.

VERSION 2 – REVIEW

REVIEWER	Ann Moyer Mayo Clinic, USA
REVIEW RETURNED	15-Jun-2019

GENERAL COMMENTS	The authors have done an excellent job of clarifying/responding to the questions raised.
--

REVIEWER	Eberechukwu Onukwugha University of Maryland School of Pharmacy
REVIEW RETURNED	11-Jun-2019

GENERAL COMMENTS	The authors have addressed my comments.
---